# Rethinking the Mixture of Vision Encoders Paradigm for Enhanced Visual Understanding in Multimodal LLMs

**Mozhgan Nasr Azadani**                                    *mnasraza@uwaterloo.ca*
*University of Waterloo*

**James Riddell**                                           *james.riddell@uwaterloo.ca*
*University of Waterloo*

**Sean Sedwards**                                           *sean.sedwards@uwaterloo.ca*
*University of Waterloo*

**Krzysztof Czarnecki**                                     *k2czarne@uwaterloo.ca*
*University of Waterloo*

**Reviewed on OpenReview:** *https://openreview.net/forum?id=tgnTVmRybs*

## Abstract

Mixture of Vision Encoders (MoVE) has emerged as a powerful approach to enhance the fine-grained visual understanding of multimodal large language models (MLLMs), improving their ability to handle tasks such as complex optical character recognition and scene understanding. Despite these advances, effectively combining diverse encoders and their visual tokens, while also scaling to high-resolution inputs, remains an open challenge. In this work, we conduct a systematic study of fusion designs for MoVE-based MLLMs, highlighting principles for token-level integration across complementary encoders. Our study shows that a lightweight recipe consisting of post-adaptation fusion with independent projectors, tile-level sequence interleaving, and dynamic tiling with global context delivers strong performance on diverse benchmarks. We integrate these principles into a simple and effective architecture that we call LEO. Extensive evaluation on 11 vision–language benchmarks demonstrates that LEO achieves better results on the majority of tasks compared to existing MoVE-based approaches. Furthermore, LEO adapts effectively to the specialized domain of autonomous driving without altering its architecture or training recipe, achieving competitive performance against established baselines and thereby highlighting its ability to generalize. The code is available at https://github.com/Mozhgan91/LEO.

## 1 Introduction

Multimodal large language models (MLLMs) (Li et al., 2023a; Alayrac et al., 2022; Liu et al., 2024c; Gao et al., 2023) have recently achieved strong performance by aligning vision encoders with large language models (LLMs) through multi-stage training on large-scale image–text datasets. This alignment allows visual tokens from pretrained vision foundation models such as CLIP Radford et al. (2021) to be mapped into the latent space of LLMs, enabling progress in a variety of vision–language reasoning tasks (Liu et al., 2024c; Driess et al., 2023). However, these models still face challenges in tasks that require fine-grained perception, such as complex optical character recognition or chart understanding, where the ability to process high-resolution inputs is critical for preserving detailed visual information. Enhancing visual understanding has therefore become a key priority, not only for improved performance in high-resolution tasks, but also for reducing hallucinations (Shi et al., 2024).

Recent studies Lin et al. (2024); Chen et al. (2025); Vasu et al. (2025); Shen et al. (2025) have explored different strategies to enhance the visual understanding capabilities of MLLMs. One line of work strengthens

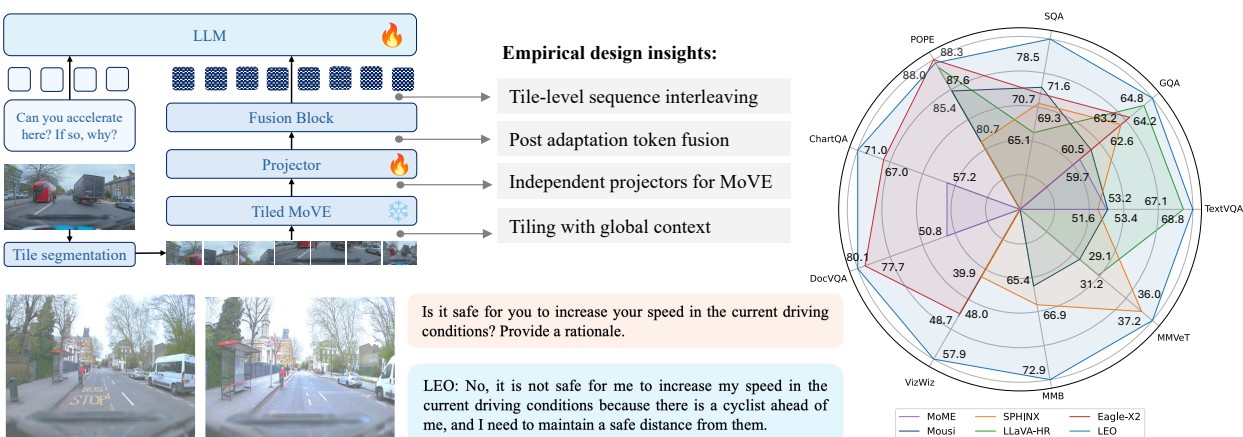

Figure 1: Overview of LEO and its performance. LEO integrates key design principles into a lightweight MoVE-based architecture for high-resolution visual reasoning.

vision encoders, either by scaling model parameters and pretraining data to match those of LLMs Zhai et al. (2023); Chen et al. (2024c); Sun et al. (2023), or by enabling fine-grained perception through tiling Liu et al. (2024b); Chen et al. (2024b); Wu et al. (2025) and high-resolution inputs Beyer et al. (2024); Zhang et al. (2024). Another promising direction is the Mixture of Vision Encoders (MoVE) paradigm, which integrates multiple pretrained experts to employ their complementary strengths. These models maintain the standard MLLM architecture while enriching visual understanding through specialized encoders. Existing MoVE-based models employ fusion strategies ranging from straightforward approaches such as sequence Kar et al. (2024); Lin et al. (2023) or channel Shi et al. (2025); Lu et al. (2024a) concatenation, and sequence interleaving Tong et al. (2024), to more advanced methods like mixture-of-resolution adaptation Luo et al. (2025), cross-attention Li et al. (2024b) or routing mechanisms Zong et al. (2024). While these methods have demonstrated gains in the visual reasoning capacity of MLLMs, most have been studied in isolation, leaving their broader interactions underexplored. As a pioneer empirical work, Eagle Shi et al. (2025) investigated encoder selection and training strategies for scaling mixtures. However, key questions remain, including how strategies for enhancing visual capacity interact, the granularity at which fusion is most effective, and whether scaling the number of experts is necessary, or if effective fusion designs can yield competitive gains.

Addressing these questions requires moving beyond isolated design choices toward a principled understanding of how MoVE models function. To this end, we conduct a systematic study to determine which strategies are most effective for integrating multiple vision encoders. Specifically, we examine three core aspects of MoVE design: how dynamic tiling interacts with MoVE; which token merging strategies are most effective, from simple concatenation to structured interleaving or cross-attention; and when fusion should be applied, either before or after adaptation into the multimodal space.

Our empirical study yields several insights into how MoVE models can be most effectively designed. We find that simple but crucial choices consistently improve multimodal reasoning. Our three key findings are as follows. **(1)** Combining the mixture of vision encoders with dynamic tiling and global context (Tiled MoVE) enables models to process high-resolution inputs without exceeding context length. This design preserves fine-grained details while strengthening overall visual understanding. **(2)** Straightforward token merging strategies often outperform more complex designs such as cross-attention Li et al. (2024b). Among these, tile-level interleaving consistently achieves the best results, surpassing both sequence append and channel concatenation. **(3)** Aligning each encoder's tokens independently with dedicated projectors before merging preserves encoder-specific features and consistently outperforms pre-adaptation fusion.

Building on these insights, we propose LEO, a lightweight and effective MoVE-based MLLM. As illustrated in Fig. 1, LEO integrates the key principles identified in our empirical study: (1) dynamic tiling with global context to preserve fine-grained details, (2) tile-level sequence interleaving to ensure efficient and balanced token integration, and (3) post-adaptation fusion with independent projectors to retain encoder-specific

strengths. Despite its simplicity, this design yields a powerful architecture that achieves strong performance across a wide range of vision–language tasks.

Our main contributions are summarized as follows:

- We conduct a systematic study of design choices in MoVE-based MLLMs, examining the interaction between visual reasoning enhancements, token-level merging strategies, and fusion timing, and identify key findings **(1)−(3)**.

- We integrate these insights into Leo, a lightweight MoVE-based MLLM that provides an efficient recipe for high-resolution visual reasoning.

- We conduct extensive experiments across multiple vision-language benchmarks, demonstrating Leo's effectiveness on the majority of tasks compared to existing MoVE models.

- We demonstrate that Leo can be applied to the specialized domain of autonomous driving without modifying its architecture or training recipe, achieving competitive results and highlighting its generalizability.

## 2 Related work

### 2.1 Multimodal large language models

With the rapid advancement of large language models (Touvron et al., 2023; Achiam et al., 2023; Team et al., 2023; Chiang et al., 2023), there has been considerable interest in multimodal extensions that enhance understanding and reasoning capabilities. Pioneering works such as BLIP-2 Li et al. (2023a); Dai et al. (2023) introduce the Q-Former to bridge the modality gap between images and text, while Flamingo Alayrac et al. (2022) enables flexible processing of mixed visual–textual sequences through a resampler for in-context few-shot learning. The LLaVA family Liu et al. (2024c;a) simplifies this design by adopting lightweight projection modules, such as linear layers or MLPs, to align vision encoder outputs with the LLM token space. Our work follows this general framework of vision encoder → alignment module → LLM, building on its simplicity and effectiveness. Rather than revisiting the alignment mechanism itself, we adopt it as a foundation and shift our focus to the integration of multiple vision encoders. In particular, we investigate principles that guide the effective fusion of complementary encoders in MoVE-based MLLMs.

### 2.2 Enhanced Visual Understanding for MLLMs

To address the constraints of lower input resolutions, recent MLLMs have concentrated on enhancing their vision encoder module. From a vision-focused standpoint, these approaches can be broadly categorized into four main strategies: (1) robust vision encoders, which design stronger backbones by scaling model size and pretraining data to better capture complex features Zhai et al. (2023); Chen et al. (2024c), (2) tile segmentation, which processes high-resolution inputs by dividing images into smaller tiles Shi et al. (2024); Liu et al. (2024c); Chen et al. (2024b); Li et al. (2024c), (3) knowledge-distilled vision encoders, where large pretrained experts are distilled into smaller encoders for efficiency while retaining fine-grained perception, as in Radio Ranzinger et al. (2024); Heinrich et al. (2025), and (4) mixture of vision encoders (MoVE), which integrates multiple vision experts into a unified backbone Lin et al. (2023); Luo et al. (2025); Tong et al. (2025); Li et al. (2024b); Shen et al. (2025); Shi et al. (2025); Wei et al. (2024).

Focusing on MoVE approaches, some models suggest merging high-resolution visual details with low-resolution tokens to enhance visual representation Luo et al. (2025); Li et al. (2024b). LLaVA-HR Luo et al. (2025) proposes a dual-pathway vision model that integrates features from high-resolution convolutional blocks with those from low-resolution ViT blocks. These pretrained vision experts can nevertheless lack key capabilities, such as text understanding and object localization. To broaden encoder capacity, several studies integrate multiple vision experts trained on diverse tasks. Brave Kar et al. (2024) and MouSi Fan et al. (2024) perform sequence appending, combining vision tokens from multiple experts into a longer sequence. Tong et al. (2024) identify distinct differences in the visual features captured by CLIP Radford et al.

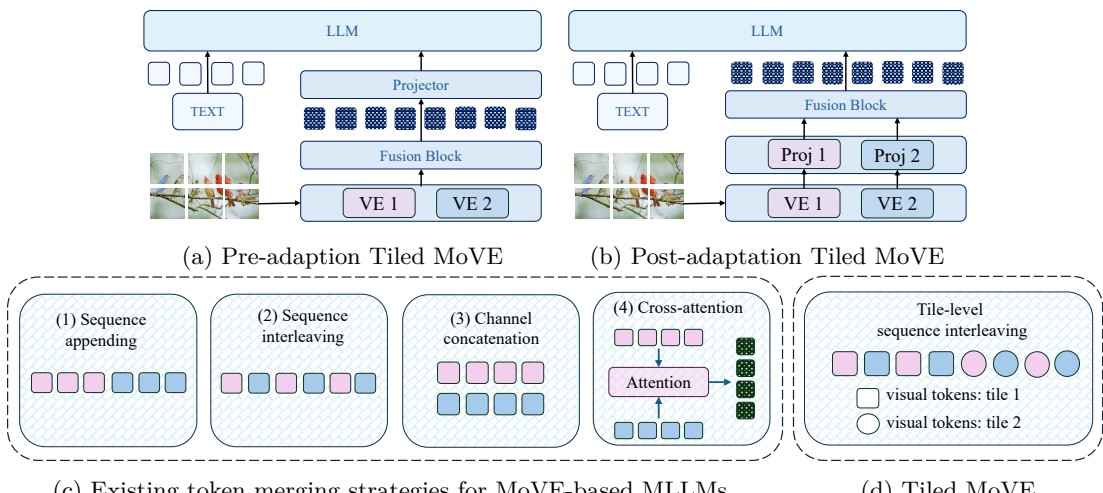

(a) Pre-adaption Tiled MoVE      (b) Post-adaptation Tiled MoVE

(c) Existing token merging strategies for MoVE-based MLLMs      (d) Tiled MoVE

Figure 2: Illustration of fusion strategies studied in our empirical analysis. (a) Pre-adaptation fusion. (b) Post-adaptation fusion. (c) The most common fusion blocks in the literature: (1) Sequence appending Kar et al. (2024), (2) Sequence interleaving Tong et al. (2024), (3) Channel concatenation Shi et al. (2025), (4) Cross-attention Li et al. (2024b),(d) A sample tile-level token merging strategy in Tiled MoVE.

(2021) and DINOv2 Oquab et al. (2024), leading to the design of an image-level mixture-of-features strategy that employs sequence interleaving. Other models apply channel concatenation to preserve sequence length, as in DeepSeek-VL Lu et al. (2024a) and Eagle Shi et al. (2025), or adopt more advanced fusion and routing mechanisms Lee et al. (2024); Zong et al. (2024).

Overall, these methods differ not only in the choice of vision encoders but also in their fusion mechanisms. Despite demonstrated gains, most have been studied in isolation with limited systematic comparisons. Our work is most related to Eagle Shi et al. (2025), which provides empirical insights into vision encoder selection and explores training strategies for scaling MoVE. Yet important questions remain: How do different enhancement strategies, such as tiling and MoVE, interact when combined? At what granularity is fusion most effective? And does stronger visual reasoning truly require scaling the number of experts? Our work is complementary. We conduct a systematic empirical study of fusion strategies for MoVE-based MLLMs. Specifically, we investigate post-adaptation fusion, sequence interleaving, and dynamic tiling with global context, offering practical guidance on which design choices are most effective.

## 3 Rethinking mixtures of vision encoders

In this section, we take a fresh look at MoVE by systematically examining the factors that govern how multiple vision encoders can be combined most effectively. In contrast to prior studies that aim to propose new merging architectures or routing mechanisms, we instead focus on deriving clear, practical insights through extensive ablations. To this end, we organize our study around three investigative directions: **D1** – how the integration of visual reasoning enhancement techniques operates; **D2** – token merging strategies, contrasting straightforward concatenation or sequence interleaving with more advanced approaches such as cross attention; and **D3** – the timing of token merging, asking whether encoder outputs are better merged prior to or following alignment. Through these investigations, we establish guiding principles that inform the design of Leo, our simple and effective MoVE-based MLLM.

### 3.1 Tiled MoVE (D1)

Recent MLLMs enhance visual understanding either by employing tile segmentation to support high-resolution inputs with a single vision encoder (Li et al., 2024c; Chen et al., 2024b), or by using a mixture of vision encoders to exploit complementary expertise (Shi et al., 2025; Kar et al., 2024; Shen et al., 2025; Lin et al., 2023; Lu et al., 2024a). While both directions have shown benefits independently, their combination

Table 1: Tiled MoVE: Investigating the impact of tiling within MoVE. *I*, *S*, *VQA$^T$*, and *MMB* denote InternViT, SigLIP, TextVQA, and MMBench, respectively.

| MoVE | Tiling Method | VQA$^T$ | GQA | VizWiz | MMB | POPE | SEED | SQA | MMVeT | Avg. |
|---|---|---|---|---|---|---|---|---|---|---|
| I + SAM | no-tiling | 62.3 | 63.1 | 52.9 | 65.1 | 86.7 | 68.8 | 68.3 | 34.1 | 62.7 |
| | fixed-grid | 62.9 | 63.0 | 53.8 | 67.5 | 87.0 | 69.4 | 71.2 | 34.2 | 63.6 |
| | overlapping | 62.6 | 63.3 | 53.4 | 66.3 | 85.6 | 69.8 | 67.3 | 33.9 | 62.8 |
| | dynamic | 63.4 | 63.9 | 54.2 | 67.5 | 87.1 | 70.8 | 71.2 | 34.6 | **64.1** |
| I + ConvNeXt | no-tiling | 61.9 | 63.4 | 52.7 | 64.9 | 86.8 | 66.3 | 65.5 | 33.7 | 61.9 |
| | fixed-grid | 62.5 | 63.5 | 53.9 | 66.0 | 86.9 | 68.1 | 69.5 | 34.3 | 63.1 |
| | overlapping | 61.9 | 62.9 | 53.6 | 65.5 | 86.9 | 66.9 | 68.1 | 33.9 | 62.5 |
| | dynamic | 62.6 | 63.9 | 54.0 | 66.4 | 87.0 | 68.5 | 71.1 | 34.6 | **63.5** |
| I + DINOv2 | no-tiling | 62.2 | 63.5 | 52.7 | 65.0 | 86.8 | 68.5 | 67.3 | 34.0 | 62.5 |
| | fixed-grid | 63.7 | 63.8 | 54.0 | 66.5 | 87.1 | 68.5 | 70.1 | 34.2 | 63.5 |
| | overlapping | 62.5 | 63.6 | 53.4 | 65.1 | 86.8 | 68.2 | 68.7 | 33.8 | 62.8 |
| | dynamic | 63.3 | 64.0 | 53.9 | 67.1 | 87.2 | 68.4 | 71.0 | 34.5 | **63.7** |
| S + SAM | no-tiling | 62.1 | 62.9 | 53.2 | 64.8 | 86.3 | 67.6 | 66.5 | 34.2 | 62.2 |
| | fixed-grid | 62.7 | 62.9 | 53.7 | 65.5 | 86.7 | 68.8 | 68.4 | 34.7 | 62.9 |
| | overlapping | 63.1 | 63.2 | 53.6 | 66.7 | 86.5 | 69.2 | 68.8 | 34.7 | 63.2 |
| | dynamic | 63.4 | 63.2 | 54.0 | 67.3 | 86.9 | 69.8 | 70.4 | 34.8 | **63.7** |
| S + ConvNeXt | no-tiling | 60.8 | 63.2 | 52.8 | 64.7 | 86.4 | 66.9 | 66.0 | 33.4 | 61.8 |
| | fixed-grid | 61.9 | 63.7 | 53.4 | 66.1 | 86.7 | 68.1 | 68.9 | 33.9 | 62.8 |
| | overlapping | 61.5 | 63.4 | 53.1 | 65.7 | 86.5 | 67.5 | 68.3 | 33.6 | 62.5 |
| | dynamic | 62.3 | 63.8 | 53.6 | 66.5 | 86.9 | 68.4 | 70.8 | 33.9 | **63.3** |
| S + DINOv2 | no-tiling | 60.4 | 63.2 | 52.8 | 65.2 | 86.2 | 68.6 | 67.2 | 33.9 | 62.2 |
| | fixed-grid | 62.8 | 63.9 | 53.3 | 66.0 | 86.9 | 70.1 | 68.4 | 34.7 | 63.3 |
| | overlapping | 61.7 | 63.7 | 53.1 | 65.7 | 86.9 | 69.6 | 68.6 | 34.0 | 62.9 |
| | dynamic | 63.2 | 64.3 | 53.9 | 66.9 | 87.2 | 70.5 | 70.1 | 34.7 | **63.9** |

has not been systematically studied. Here, we investigate how different tiling methods and MoVE interact when applied together.

To explore this interaction, we evaluate four tiling strategies across multiple encoder combinations as preprocessing for high-resolution inputs: (1) *No-tiling*, where the full image is treated as a single tile, (2) *Fixed-grid* tiling, which partitions the image into a uniform grid of equally sized tiles, (3) *Overlapping* tiling, which uses tiles of the same size but allows adjacent tiles to partially overlap for denser spatial coverage, and (4) *Dynamic* tiling, which adapts the number and arrangement of tiles based on the image aspect ratio while keeping the tile size fixed. These strategies differ in how they partition a high-resolution image, and therefore vary in the spatial detail they preserve and the redundancy they introduce. Given an image $I_{\text{img}} \in \mathbb{R}^{H \times W \times 3}$, each method partitions it into a set of tiles $I_{\text{tiles}} = I_1, I_2, \ldots, I_N$, where each tile $I_n \in \mathbb{R}^{(H_{\text{tile}} \times W_{\text{tile}}) \times 3}$ has a dimension of $H_{\text{tile}} \times W_{\text{tile}}$.

As an illustrative example, we describe the dynamic tiling procedure in detail. Dynamic tiling Chen et al. (2024b) is aspect-ratio–aware, adapting the number and shape of tiles to the image geometry. The input image $I_{\text{img}}$ is resized to a closest available aspect ratio that is dividable into square patches of size $448 \times 448$. For example, for common aspect ratios such as 3:2, the procedure produces up to six tiles, ensuring coverage of informative regions (see Fig. 3). In parallel, we generate a thumbnail representation $I_t$ of the full image, which preserves global structure and provides complementary context to the localized tile features. Figure 3 illustrates this tiling process with an example driving scene image, where the tiles are shown after normalization by the SAM preprocessor Kirillov et al. (2023).

Each tile is then processed independently by two pretrained vision encoders, yielding embeddings $I_n^{v_1}$ and $I_n^{v_2}$. Because these encoders are trained on different domain-specific vision tasks and objectives, they emphasize distinct aspects of the visual signal; one may specialize in semantic categorization, while another better captures fine-grained textures or geometric cues. This diversity allows the downstream fusion module to

benefit from complementary perspectives of the same visual input. To reduce the number of visual tokens and ensure that both vision encoders generate the same number of tokens, we apply pixel unshuffling Shi et al. (2016) if necessary. This technique rearranges the spatial layout of pixels, reducing the number of visual tokens while preserving important visual features. Given an input visual embedding $I_n^{v_1} \in \mathbb{R}^{C_1 \times (V_1 = h_1 \times w_1)}$, and a downscaling factor $r$, the module outputs $\bar{I}_n^{v_1} \in \mathbb{R}^{(C_1 \times (r^2)) \times (h_1/r \times w_1/r)}$. Finally, each segmented tile is represented by 256 visual tokens per encoder. Experimental settings are provided in Appendix A.2.

Table 1 reports the effect of incorporating different tiling strategies into MoVE across a variety of vision encoders, including InternViT Chen et al. (2024c), SAM Kirillov et al. (2023), SigLIP Zhai et al. (2023), ConvNeXt Woo et al. (2023), and DINOv2 Oquab et al. (2024). Across all configurations, we observe three consistent patterns. First, dynamic tiling achieves the strongest performance, improving the average score in all encoder combinations. For example, pairing InternViT with SAM, ConvNeXt, or DINOv2 yields clear gains of 2.3%, 2.6%, and 1.9%, respectively, compared to the no-tiling baseline. This illustrates the benefit of adapting the number of tiles to the image aspect ratio, allowing

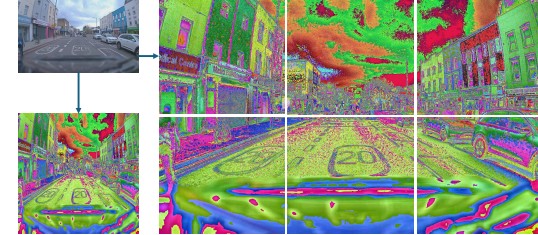

Figure 3: Dynamic tiling with global context. The tiles are shown after preprocessing with the SAM preprocessor Kirillov et al. (2023).

the model to preserve spatial structure while maintaining a stable token budget. Second, fixed-grid tiling typically ranks second, outperforming both the no-tiling and overlapping settings. Its uniform grid increases spatial coverage, although the inability to adapt to diverse image geometries may limit its effectiveness relative to dynamic tiling. Third, overlapping tiling provides only modest gains and often remains close to the no-tiling baseline, suggesting that the redundant coverage introduced by overlapping regions does not yield proportionally more informative tokens for the LLM. Overall, these results demonstrate that tiling and MoVE are complementary: while MoVE uses the diversity of multiple vision encoders, tiling strengthens each encoder's ability to process high-resolution visual content, leading to more robust multimodal understanding.

## 3.2 Token merging strategies for Tiled MoVE (D2)

Existing MLLM frameworks have explored different ways of merging tokens from multiple vision encoders, aiming to exploit their complementary strengths. While improvements in multimodal performance have been consistently reported, the contribution of the merging strategy itself has rarely been isolated. In most cases, token merging is entangled with other architectural innovations, making it difficult to assess whether gains arise from the fusion design or from stronger encoder representations. Shi et al. (2025) investigated MoVE fusion strategies with respect to the trade-off between accuracy and efficiency, but their analysis focused on image-level fusion in general. In contrast, our work examines token merging strategies at the tile level (see Fig 2d): within Tiled MoVE, we perform a controlled comparison of four representative methods under a consistent tiled MoVE setting.

Figure 2c illustrates the four most common token merging strategies used in recent MoVE-based MLLMs. In this work, we explore these strategies to combine token sequences from two vision encoders. (1) *Sequence Appending* (SA) Kar et al. (2024), where tokens from each encoder are concatenated along the sequence dimension. If a tile produces $D_c$ tokens per encoder, the combined sequence has length $2D_c$. (2) *Sequence Interleaving* (SI) Tong et al. (2024), where tokens are interleaved position by position, e.g., $[t_1^{v_1}, t_1^{v_2}, \ldots, t_{D_c}^{v_1}, t_{D_c}^{v_2}]$, which

Table 2: Comparison of various token merging strategies within Tiled MoVE across 8 benchmarks (see Appendix A.3).

| MoVE | Merging | Avg. (8 benchmarks) |
|---|---|---|
| I + SAM | SA | 64.3 |
| | SI | **64.9** |
| | CC | 64.1 |
| | CA | 63.5 |
| I + ConvNeXt | SA | **63.7** |
| | SI | **63.7** |
| | CC | 63.5 |
| | CA | 62.5 |
| I + DINOv2 | SA | 65.1 |
| | SI | **66.7** |
| | CC | 63.7 |
| | CA | 62.9 |
| S + SAM | SA | 64.2 |
| | SI | **64.6** |
| | CC | 63.7 |
| | CA | 63.0 |
| S + ConvNeXt | SA | 63.5 |
| | SI | **64.0** |
| | CC | 63.3 |
| | CA | 62.5 |
| S + DINOv2 | SA | 64.8 |
| | SI | **66.4** |
| | CC | 63.9 |
| | CA | 62.5 |

preserves the order while mixing encoder streams. As above, the combined sequence has length $2D_c$. (3) *Channel Concatenation* (CC) Shi et al. (2025), where instead of extending the sequence length, the features from both encoders at each token position are concatenated along the channel dimension, resulting in vectors of size $C_1 + C_2$ per token. (4) *Cross-Attention* (CA) Li et al. (2024b), where tokens from one encoder act as queries attending to the other encoder's keys and values, enabling adaptive integration of complementary information through cross-attention.

Table 2 summarizes the effect of different token merging strategies within Tiled MoVE across six encoder pairs. We observe that sequence interleaving achieves the best performance in five out of six combinations, with the only exception being *I + ConvNeXt*, where it ties with sequence append. We hypothesize that interleaving the visual tokens at the tile level helps preserve spatial relationships while improving information integration, leading to stronger overall performance. In contrast, channel concatenation and cross-attention consistently underperform relative to interleaving and appending. Overall, these results highlight the importance of the token merging strategy, with interleaving emerging as the most effective design choice for Tiled MoVE. An efficiency analysis of the four token-merging strategies revealed distinct accuracy–throughput trade-offs (See Appendix A.5).

### 3.3 Pre-Adaptation versus Post-Adaptation Fusion Strategy (D3)

An important design consideration in MoVE models is *when* fusion should occur: should visual tokens be merged immediately after feature extraction, or only after each encoder has been adapted to the multimodal backbone? Most existing MoVE MLLMs Luo et al. (2025); Li et al. (2024b); Kar et al. (2024); Shi et al. (2025); Fan et al. (2024); Lu et al. (2024a) adopt the former, reffered to as *pre-adaptation* fusion, where tokens from different encoders are merged before the vision-text alignment stage. In this setup, visual tokens from different encoders are first merged (by a standard token merging method), and the fused tokens are then mapped into the multimodal space using a single shared projector module.

In contrast, *post-adaptation* fusion equips each encoder with its own dedicated projector, such that visual embeddings are first aligned independently before being merged. For instance, the outputs $\bar{I}_n^{v_1}$ and $\bar{I}_n^{v_2}$ from two encoders are mapped through their respective projectors, $\Gamma_{I \to T_1}(\bar{I}_n^{v_1}) \to T_n^{v_1}$ and $\Gamma_{I \to T_2}(\bar{I}_n^{v_2}) \to T_n^{v_2}$, producing normalized token sequences prior to fusion. This design allows the integration step to operate on representations that are already aligned with the multimodal backbone while preserving encoder-specific characteristics. The relative effectiveness of these two strategies has not yet been systematically investigated in the literature. Accordingly, we design a controlled empirical study within Tiled MoVE to isolate the effect of fusion timing by contrasting pre- and post-adaptation approaches. We adopt a simple and effective two-layer MLP as the projector design. Each vision encoder output, $\bar{I}_n^{v_1}$ and $\bar{I}_n^{v_2}$, is independently mapped into aligned token sequences $T_n^{v_1}, T_n^{v_2} \in \mathbb{R}^{D_c \times D_s}$, where $D_c$ is the per-tile token length and $D_s$ matches the hidden dimension of the LLM. This ensures compatibility with the multimodal backbone while maintaining encoder-specific characteristics. Experimental settings are provided in Appendix A.2.

Table 3 presents the comparison between pre- and post-adaptation fusion across several encoder combinations. In all cases, post-adaptation yields consistent improvements over pre-adaptation, highlighting the benefits of aligning encoder outputs independently before fusion. On average, post-adaptation delivers gains of around 2.9% across the evaluated backbones. We hypothesize that these improvements stem from allowing each encoder to independently align its features before fusion, which both preserves encoder-specific information and ensures that the subsequent fusion operates on already normalized representations. Moreover, the consistency of these gains across five different encoder pairs suggests that preserving encoder-specific features during alignment facilitates stronger integration between visual and language representations, ultimately leading to better multimodal reasoning and overall performance. We also compare the computational efficiency of pre- and post-adaptation fusion for the *I + SAM* encoder pair. The latency difference between the two designs is minimal, with post-adaptation showing a slight advantage (3.19s vs. 3.25s). This indicates that both approaches are computationally comparable, with post-adaptation providing marginally faster generation while achieving higher accuracy.

Table 3: Comparison of pre-adaptation vs. post-adaptation fusion strategies within Tiled MoVE. *I*, $VQA^T$, and *MMB* denote InternViT, TextVQA, and MMBench, respectively.

| MoVE | Fusion Method | VQA$^T$ | GQA | VizWiz | MMB | POPE | SEED | SQA | MMVeT | Avg. |
|---|---|---|---|---|---|---|---|---|---|---|
| I + SAM | pre-adaptation | 65.2 | 64.2 | 54.7 | 68.7 | 87.8 | 69.1 | 74.6 | 35.2 | 64.9 |
| | post-adaptation | 68.8 | 64.8 | 57.9 | 72.9 | 88.0 | 72.2 | 78.5 | 37.2 | **67.5** |
| I + SigLIP | pre-adaptation | 69.2 | 62.4 | 53.6 | 67.8 | 86.5 | 71.2 | 72.3 | 33.0 | 64.5 |
| | post-adaptation | 70.4 | 64.8 | 56.5 | 70.5 | 88.2 | 73.0 | 74.5 | 36.9 | **66.8** |
| I + ConvNeXt | pre-adaptation | 64.2 | 63.3 | 54.7 | 64.7 | 86.4 | 68.3 | 73.4 | 34.6 | 63.7 |
| | post-adaptation | 67.3 | 63.7 | 56.7 | 66.8 | 87.4 | 72.0 | 75.4 | 35.2 | **65.6** |
| I + CLIP | pre-adaptation | 65.0 | 62.5 | 48.4 | 68.9 | 86.4 | 71.6 | 70.4 | 31.3 | 63.1 |
| | post-adaptation | 67.0 | 63.8 | 48.4 | 72.7 | 87.8 | 73.0 | 75.7 | 34.3 | **65.3** |
| I + DINOv2 | pre-adaptation | 68.5 | 64.3 | 56.8 | 71.3 | 88.0 | 72.9 | 76.8 | 35.2 | 66.7 |
| | post-adaptation | 70.6 | 64.8 | 57.3 | 72.7 | 87.8 | 73.2 | 74.3 | 35.4 | **67.0** |

# 4 LEO

Having systematically analyzed tiling (D1), token merging strategies (D2), and the timing of fusion (D3), we now combine these insights into a unified multimodal large language model, which we call LEO. This section introduces the overall design of LEO, constructed with the best-performing settings identified in our empirical study: dynamic tiling for high-resolution processing, sequence interleaving for effective token merging, and post-adaptation fusion for stronger alignment across encoders. In this Section, we describe the model's architecture in detail and further discuss how LEO can be adapted to autonomous driving scenarios, where fine-grained perception and robust reasoning are both critical.

LEO begins by dividing an input image $I_{\text{img}} \in \mathbb{R}^{H \times W \times 3}$ into a set of tiles $I_{\text{tiles}} = \{I_1, I_2, \ldots, I_N\}$, along the lines of Chen et al. (2024a), where each tile $I_n \in \mathbb{R}^{H_{\text{tile}} \times W_{\text{tile}} \times 3}$ captures a high-resolution patch of the scene. This tiling step preserves fine-grained details without exceeding the token budget.

Each tile is processed by two complementary vision encoders, $VE_1$ and $VE_2$. In practice, our implementation follows a ViT–projector–LLM pipeline, using InternViT-300M Chen et al. (2024c) as the first encoder, selected for its strong vision–language alignment and SAM-L Kirillov et al. (2023) as the second ecnoder, chosen for its ability to capture segmentation-based, region-level features (see Section 3). Architecturally, LEO does not impose any explicit weighting or priority on tokens from either encoder.

The two encoders extract embeddings $I_n^{v_1} \in \mathbb{R}^{C_1 \times V_1}$ and $I_n^{v_2} \in \mathbb{R}^{C_2 \times V_2}$, which are compressed through pixel unshuffling to yield compact representations $\bar{I}_n^{v_1}$ and $\bar{I}_n^{v_2}$. These are projected into the multimodal space using two independent MLP-based projectors, $\Gamma_{I \to T_1}(\bar{I}_n^{v_1}) \to T_n^{v_1}$ and $\Gamma_{I \to T_2}(\bar{I}_n^{v_2}) \to T_n^{v_2}$, where $T_n^{v_1}, T_n^{v_2} \in \mathbb{R}^{D_c \times D_s}$ are aligned token sequences. This post-adaptation design ensures that each encoder is independently normalized before fusion, preserving encoder-specific information.

For fusion, LEO employs a tile-level sequence interleaving strategy $F(T_n^{v_1}, T_n^{v_2}) \to T_v$, which merges tokens from the two encoders in alternating order within each tile. This design maintains local spatial structure while promoting cross-encoder interaction. The fused visual tokens $T_v$ are then combined with text tokens $T_t$ and processed by the multimodal backbone LLM $\Phi(T_v, T_t)$ for joint vision–language reasoning:

$$p(Y \mid T_v, T_t) = \prod_{i=1}^{L} p(y_i \mid T_v, T_t, y_{<i}). \tag{1}$$

## 4.1 Adaption to autonomous driving

Although numerous studies have successfully applied MLLMs to autonomous driving (Marcu et al., 2024; Cao et al., 2024; Tian et al., 2024; Wang et al., 2024), a straightforward approach that avoids extensive modifications to model architecture, training processes, or heavy data collection has yet to be fully explored.

In this work, we investigate the potential of applying Leo to the autonomous driving domain without altering its architecture or training recipe, aiming to offer insights into streamlined transfer learning and facilitate MLLM adaptation to specialized domains. Instruction tuning plays a crucial role in helping models learn to follow user prompts, utilizing training data in visual question answering and conversational formats. For this domain, we design tasks in a VQA format, with each frame represented as: *  </img>*. At the prompt level, the temporal aspect of video frames is managed by treating sequential frames as multiple image inputs. A sample prompt is formulated as "*<image1> ... <image N> Is it safe to enter the intersection at this time?*".

## 5 Experiments

### 5.1 Implementation Details.

**Training Procedure.** The training of Leo is performed in two stages. The first stage serves as a warm-up phase for the projector layers, where both vision encoders remain frozen and optimization focuses solely on the projector modules to ensure stable and effective alignment. To mitigate representation inconsistencies, the initialization of the projector MLPs depends on the encoder type: if the encoder has been pretrained on a vision–language task, we initialize its projector from pretrained weights; otherwise, we initialize it randomly. The second stage involves supervised instruction tuning, where the projector modules are unfrozen together with the language model. Both stages are trained with a context length of 8196 tokens using the AdamW optimizer and a cosine learning rate schedule. In the first stage, we apply a learning rate of $4 \times 10^{-4}$ with a weight decay of 0.01, while in the second stage the learning rate is reduced to $4 \times 10^{-5}$ with the same weight decay. Each stage is trained for one epoch.

**Training Datasets.** For the first (alignment) stage, we use the LLaVA-595K dataset Liu et al. (2024c), which contains 595K vision-language instruction pairs. For the second (supervised fine-tuning) stage, we follow the same dataset setup as the baseline model Chen et al. (2024c), comprising approximately 1M visual instruction tuning samples, all publicly available.

**Training Infrastructure.** Training was performed on 8 NVIDIA A100 GPUs (80 GB each) using Deep-Speed's ZeRO-2 optimization strategy. The complete training process took approximately 72 hours.

**Benchmarks.** We evaluate Leo on a broad suite of multimodal benchmarks, grouped into three categories. (1) *OCR and chart understanding*: DocVQA Mathew et al. (2021), TextVQA Singh et al. (2019), ChartQA Masry et al. (2022), and AI2D Kembhavi et al. (2016). (2) *General VQA*: GQA Hudson & Manning (2019), VizWiz Gurari et al. (2018), and VQA v2 Goyal et al. (2017). (3) *General multimodal evaluation*: MMMU Yue et al. (2024), MMBench Liu et al. (2025), SEED Li et al. (2024a), POPE Li et al. (2023b), MM-Vet Yu et al. (2024), and ScienceQA Lu et al. (2022), MathVista Lu et al. (2024b). In addition, we assess Leo in the autonomous driving domain using LingoQA Marcu et al. (2024), which requires reasoning about road scenes and language-based driving queries. Further details are provided in Appendix A.1.

### 5.2 Main Results

Table 4 presents a comprehensive comparison of Leo with existing MoVE-based MLLMs across 11 benchmarks. Overall, Leo delivers strong results in 7 out of 11 tasks, demonstrating the effectiveness of the design principles established through our empirical analysis. In particular, Leo achieves strong performance on DocVQA (80.1) and ScienceQA (78.5), with margins of improvement that are substantially greater than those observed in prior models, indicating robust capabilities in both text-heavy OCR settings and reasoning-oriented benchmarks. Leo also records clear gains on ChartQA (+3.2%) and VizWiz (+3.7%), showing that the proposed architecture can generalize effectively to both structured visual data and challenging real-world images. On MMBench (72.9), Leo remains competitive, ranking just below the top-performing system. Importantly, these improvements are achieved with substantially less pretraining and instruction-tuning data than models such as SPHINX Lin et al. (2023) and DeepSeek-VL Fan et al. (2024), demonstrating that Leo's performance gains stem from architectural design rather than data scale.

Table 4: Comparison with MoVE MLLMs on 11 evaluation benchmarks. All models use a 7B language model. The best and second best values are shown in **bold** and underlined, respectively.

| Model | PT | SFT | TextVQA | GQA | ChartQA | VizWiz | MMBench | MMVet | DocVQA | POPE | SEED | ScienceQA | MathVista |
|---|---|---|---|---|---|---|---|---|---|---|---|---|---|
| Brave-X5 Kar et al. (2024) | 100 M | NA | - | 52.7 | - | 54.2 | - | - | - | 87.6 | - | - | - |
| Deepseek-VL Fan et al. (2024) | 3.7 M | 3 M | - | - | - | - | **73.2** | **41.5** | - | 88.1 | 70.4 | - | 36.1 |
| Eagle-X2 Shi et al. (2025) | 595 K | 1.8 M | - | 63.2 | 67.0 | 48 | - | - | 77.7 | 88.3 | 73.5 | 70.7 | - |
| Eagle-X3 Shi et al. (2025) | 595 K | 1.8 M | - | 63.2 | 67.8 | 51.7 | - | - | 77.7 | **89** | **73.9** | 69.4 | - |
| LLaVA-HR Luo et al. (2025) | 558 K | 1.2 M | 67.1 | 64.2 | - | 48.7 | - | 31.2 | - | 87.6 | 64.2 | 65.1 | - |
| Mini-Gemini Li et al. (2024b) | 1.2 M | 1.5 M | 65.2 | - | - | - | 69.3 | 40.8 | - | - | - | 71.1 | 31.4 |
| MoME-X3 Shen et al. (2025) | NA | NA | 53.2 | 59.7 | 57.2 | - | - | - | 50.8 | - | - | - | - |
| MouSi-X2 Fan et al. (2024) | 1.2 M | 1.6 M | 53.4 | 60.5 | - | - | 65.4 | 29.1 | - | 85.4 | 62.0 | 71.6 | - |
| MouSi-X3 Fan et al. (2024) | 1.2 M | 1.6 M | 58 | 63.3 | - | - | 66.8 | 32.2 | - | 87.3 | 66 | 70.2 | - |
| SPHINX Lin et al. (2023) | 400 M | NA | 51.6 | 62.6 | - | 39.9 | 66.9 | 36.0 | - | 80.7 | 69.1 | 69.3 | 27 |
| LEO | 595 K | 1 M | **68.8** | **64.8** | **71.0** | **57.9** | 72.9 | 37.2 | **80.1** | 88.0 | 72.2 | **78.5** | **37.2** |
| Δ | - | - | ↑ 1.7 | ↑0.6 | ↑3.2 | ↑3.7 | ↓0.3 | ↓4.3 | ↑2.4 | ↓0.9 | ↓1.7 | ↑6.9 | ↑1.1 |

A closer comparison with strong baselines further highlights the efficiency of our approach. Models such as LLaVA-HR Luo et al. (2025) and Mini-Gemini Li et al. (2024b), which employ more complex fusion mechanisms, still lag behind on reasoning-intensive tasks including SEED and ScienceQA. Likewise, Brave Kar et al. (2024), despite integrating as many as five vision encoders through pre-adaptation fusion, achieves weaker or only comparable results. LEO attains competitive or superior performance with just two encoders. These findings suggest that the consistent improvements of LEO arise from the design principles identified in our empirical study, including tiling for high-resolution processing, sequence interleaving for balanced token integration, and post-adaptation fusion for preserving encoder-specific strengths. Taken together, these results establish LEO as both a data- and computation-efficient multimodal model, capable of outperforming or matching more heavily engineered MoVE MLLMs while using a straightforward and effective architecture.

## 5.3 Ablation studies

**Analysis of training strategies.** To more effectively analyze the impact of training strategies for the vision encoders, we conduct ablation studies to test whether unfreezing the vision backbones improves performance (Table 5). We first observe that using a single encoder alone results in reduced performance: InternViT by itself achieves a reasonable average score of 60.8, while SAM alone performs much worse (51.4), reflecting its specialization in segmentation rather than general visual–language understanding. Freezing SAM slightly improves stability and yields a small gain (53.7), but it still lags far behind InternViT. Combining InternViT and SAM without freezing substantially boosts performance to 65.6, confirming that their complementary strengths—InternViT's strong vision–language alignment and SAM's fine-grained region cues—synergize when trained together.

Interestingly, the results in Table 5 show that the best performance is achieved when both vision encoders are frozen at SFT, reaching an average of 67.5. This finding stands in sharp contrast to Eagle Shi et al. (2025), where unfreezing the encoders was found to be necessary for competitive results. We hypothesize that this difference stems from the tiling setup in Tiled MoVE: since each encoder already processes high-resolution patches, their pretrained representations remain highly effective without further adapta-

Table 5: Ablation study on training settings for vision backbones (full results in Appendix A.3).

| InternViT | SAM | Freeze | Avg. (8 benchmarks) |
|---|---|---|---|
| ✓ | ✗ | ✗ | 60.8 |
| ✗ | ✓ | ✗ | 51.4 |
| ✗ | ✓ | ✓ | 53.7 |
| ✓ | ✓ | ✗ | 65.6 |
| ✓ | ✓ | ✓ | **67.5** |

tion. Freezing prevents catastrophic forgetting of these pretrained priors and shifts the burden of adaptation to the lightweight projector modules, which can more efficiently normalize encoder-specific embeddings into the multimodal space. This design leads to more stable training and better overall reasoning performance.

**Analysis of fusion strategies.** To further isolate the contribution of individual design choices in Leo, we conduct an ablation across eight benchmarks, with average results summarized in Table 6. While Sections 3.2 and 3.3 examined token merging and fusion strategies more broadly, here we focus on their impact within the finalized Leo setting. We find that tile-level sequence interleaving yields the best performance (avg. 67.5), surpassing both sequence appending (avg. 66.5) and channel concatenation (avg. 66.1). This confirms the advantages of interleaving in balancing token integration across encoders. In addition, removing dynamic tiling leads to a clear drop in per-

Table 6: Ablation study on various types of fusion strategies and SFT data (full results in Appendix A.3).

| Model | Avg. (8 benchmarks) |
|---|---|
| Leo | **67.5** |
| w/ pre-adaptation | 64.9 |
| w/ sequence appending | 66.5 |
| w/ channel concatenation | 66.1 |
| w/o tile segmentation | 64.6 |
| w/ 1.8M SFT data | 67.3 |

formance (avg. 64.6), underscoring the role of high-resolution tiling in preserving fine-grained details for visual understanding. Finally, post-adaptation fusion again outperforms pre-adaptation, reinforcing that aligning encoder outputs independently before fusion facilitates stronger integration with the multimodal backbone. Together, these results echo the broader analyses from Sections 3.2 and 3.3 while providing a controlled validation of Leo's design principles.

**Effect of training data scale.** We evaluate the impact of increasing SFT data by training Leo with Eagle-1.8M SFT data Shi et al. (2025). While this leads to improvements on three benchmarks (see Appendix A.3), the overall average remains stable (67.3 vs. 67.5), suggesting that Leo achieves strong performance even with limited data, demonstrating its training efficiency and robustness.

### 5.4 How Does LEO Compare to Single-Encoder Models with Knowledge Distillation?

To further contextualize Leo 's performance, Table 7 compares it against models that employ a single vision encoder obtained through multi-teacher knowledge distillation Ranzinger et al. (2024); Heinrich et al. (2025); Cao et al. (2025). These approaches compress information from several teacher encoders into a single

Table 7: Comparison with knowledge distillation (multiple vision encoder teachers) approaches.

| | GQA | ChartQA | VQA$^T$ | DocVQA | POPE |
|---|---|---|---|---|---|
| RADIO Ranzinger et al. (2024) | 63.01 | - | 56.32 | - | 86.20 |
| RADIO2.5 Heinrich et al. (2025) | - | 30.40 | **69.74** | 52.33 | - |
| MoVE-KD-v1.1 Cao et al. (2025) | 63.9 | - | 59.6 | - | 86.3 |
| Leo | **64.8** | **71.0** | 68.8 | **80.1** | **88.0** |

backbone during training. For example, RADIO uses per-teacher adapters only during the distillation stage to better align teacher features; for the VQA task at inference, the trained student model operates as a standard single vision encoder followed by a two-layer MLP adaptor. Consistent with their original design, these models are evaluated using their default single-tile input resolution. Despite relying on substantially greater pretraining (RADIO is trained on 614M samples with 64 GPUs, and RADIO-2.5 on 9.8M samples), Leo achieves competitive or stronger performance across most benchmarks using only 1M training samples and 8 GPUs. It is worth mentioning that knowledge-distilled encoders such as RADIO or MoVE-KD's encoder have lower inference-time latency than MoVE encoders, since they require only a single backbone forward pass, whereas MoVE-based approaches perform one forward pass per encoder. Furthermore, unlike knowledge distillation-based models, which blur the representational roles of individual vision experts, Leo preserves modularity and interpretability by explicitly integrating multiple encoders. Moreover, Leo shows equally strong performance against general MLLMs with similar resource constraints (See Appendix A.4).

### 5.5 Efficiency analysis

Table 9 compares the efficiency of Leo and Eagle Shi et al. (2025) in terms of vision encoder parameters, latency, FLOPs, and generation time, using a 1024×1024 input image with the prompt *"describe this image in detail"*. Leo demonstrates significant efficiency improvements with only 612M vision encoder parameters—just over half the parameters of Eagle-X2 and less than half those of Eagle-X3. Despite higher latency due to SAM's lack of flash attention support Dao (2024), Leo achieves a notable 61.6% reduction in vision encoder FLOPs and a 19.6% decrease in generation time compared to Eagle-X3, highlighting its efficiency.

## 5.6 Results in the autonomous driving domain

We evaluate LEO on the LingoQA validation set Marcu et al. (2024), with results presented in Table 8. Against the closed-source LingoQA baseline Marcu et al. (2024), which is pretrained on over 22M data samples, LEO demonstrates competitive performance on the Lingo-J and BLUE metrics, and significantly outperforms the baseline on the METEOR and CIDEr metrics. Without modifying its architecture or training recipe, LEO also surpasses all existing open-source baselines across all four

Table 8: Results on the LingoQA benchmark Marcu et al. (2024). *N* denotes the number of frames used during training. *Lingo-J* represents the Lingo-Judge metric.

| Model | N | Lingo-J ↑ | BLUE ↑ | METEOR ↑ | CIDEr ↑ |
|---|---|---|---|---|---|
| BLIP 2 Li et al. (2023a) | 1 | 52.20 | 13.00 | 17.40 | 60.10 |
| LLaVA-1.5 Liu et al. (2024a) | 5 | 51.00 | 10.62 | 29.44 | 48.18 |
| InternVL Chen et al. (2024c) | 5 | 58.00 | 13.53 | 34.27 | 67.17 |
| LingoQA Marcu et al. (2024) | 3 | 59.80 | 14.61 | 18.44 | 62.61 |
| LingoQA Marcu et al. (2024) | 5 | 60.80 | **15.00** | 18.56 | 65.62 |
| LEO (ours) | 2 | **61.00** | 14.91 | **35.44** | **69.72** |

metrics. Notably, LEO achieves higher scores than the top-performing model of Chen et al. (2024c). These results indicate that LEO can better capture fine-grained multimodal cues, a critical property for tasks such as scene understanding and instruction following in dynamic driving environments.

## 5.7 Visualization

To highlight the visual understanding capabilities of LEO, we present a qualitative analysis in Fig. 4. Our model is applied to a variety of vision-language tasks, including complex reasoning, detailed counting, OCR, spatial and mathematical reasoning, accounting analysis, and multi-image and multi-frame reasoning. With an efficient tile-level post-adaptation fusion strategy, LEO exhibits impressive

Table 9: Efficiency analysis of LEO. 'GT' and 'lat' denote generation time and latency, respectively.

| Model | VE Param. | VE lat. | VE FLOPs | GT |
|---|---|---|---|---|
| Eagle-X3 | 1460M | 0.07s | 4281 | 3.97s |
| Eagle-X2 | 1155M | 0.04s | 3347 | 3.92s |
| LEO | 612M | 0.78s | 1642 | 3.19s |

performance across these challenging tasks. For example, our model can perform attribute-based counting, such as identifying the absence of parked cars while there are several moving vehicles in the driving scene. Beyond simple recognition, LEO demonstrates spatial awareness, enabling it to answer OCR-related questions like "*What is located to the right of the shampoo?*" In multi-image reasoning, LEO accurately identifies detail differences between images, such as the dog's head being in different positions. LEO also demonstrates strong capabilities in multi-frame reasoning in the autonomous driving domain, including recognizing safe actions in dynamic scenes, such as stopping to allow a pedestrian to cross. Finally, LEO excels in OCR tasks, effectively interpreting dense text, and also handles complex mathematical and accounting problems, showcasing its strong reasoning abilities. See Appendix A.7.

## 5.8 Limitations

The processing capacity of our model is limited to a maximum of six image patches (excluding the global context), constrained by the LLM's context length and available computational resources. This restriction hinders support for higher-resolution images or tasks requiring a larger number of multi-image inputs. Moreover, due to computational constraints, our setup relies on a fixed set of vision encoders and a 7B LLM, which, while efficient, may limit the diversity of visual features and the breadth and depth of the system's reasoning abilities.

## 6 Conclusion

Through a systematic empirical study, we have investigated how different design choices within MoVE influence the effectiveness of multimodal large language models. We organized our study around three investigative directions: visual reasoning enhancement methodologies, token merging strategies, and the timing of fusion. From our experiments, we have derived clear principles for effectively combining multiple vision encoders. We have demonstrated that it is possible to consistently strengthen multimodal reasoning in a computationally efficient way by means of judicious lightweight design choices. Specifically, by integrating

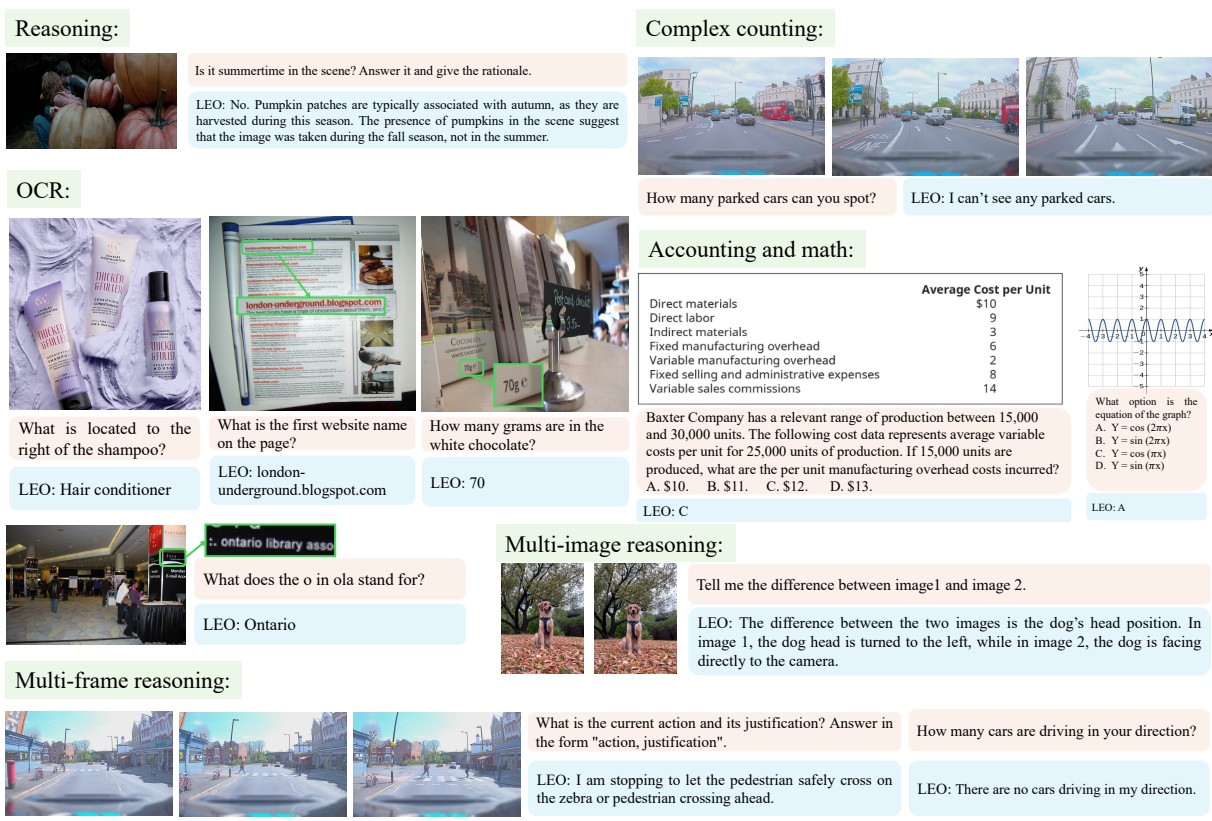

Figure 4: Qualitative results of LEO's enhanced visual understanding on various vision-language tasks. Images are taken from the following benchamrks: MMVet Yu et al. (2024), MMMU Yue et al. (2024), TextVQA Singh et al. (2019), and LingoQA Marcu et al. (2024)

dynamic tiling with MoVE, by employing tile-level sequence interleaving, and by adopting post-adaptation fusion. Using these principles, we designed a compact and powerful architecture that we call LEO. LEO achieves better results than prior MoVE models on the majority of benchmarks and is competitive with the best models on benchmarks that it does not lead. LEO further demonstrates its generalizability by transferring effectively to the specialized domain of autonomous driving without requiring extensive domain-specific adjustments. We envision LEO as a foundation for advancing MoVE-based MLLMs and as a practical guide for adapting them efficiently to domain-specific applications.

## Broader Impact Statement

The proposed LEO framework extends the capabilities of existing vision–language models and therefore shares many of their broader social impacts. As with other multimodal systems, potential risks include biased or misleading outputs, privacy concerns, and the propagation of harmful content. These risks are mitigated in part by thoughtful data selection and the application of safeguards designed to encourage safe and responsible use. We present LEO primarily as a research contribution, and recommend that its outputs be applied with caution in sensitive contexts, with human oversight to mitigate possible misuse.

## Acknowledgement

This work was supported by the Natural Sciences and Engineering Research Council of Canada (NSERC)'s Postdoctoral Fellowship and NSERC-CSE Research Community project entitled "An End-to-End Approach to Safe and Secure AI Systems". Researchers funded through the NSERC-CSE Research Communities Grants do not represent the Communications Security Establishment Canada or the Government of Canada. Any

research, opinions, or positions they produce as part of this initiative do not represent the official views of the Government of Canada.

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

# A    Appendix

A.1: Benchmark Datasets

A.2: Experimental Settings

A.3: Benchmark Details

A.4: Leo performance beyond the MoVE setting

A.5: Additional efficiency analysis

A.6: Does Scaling the Training Data Improve Cross-Attention Merging Performance?

A.7: Additional Visualization Results

## A.1    Benchmark Datasets

This section provides a more detailed description of the evaluation benchmarks, highlighting their key features. Unless specified otherwise, we report scores based on the performance of models evaluated on the provided test splits. We consider the following benchmarks:

1. OCR and chart understanding benchmarks:

   **DocVQA Mathew et al. (2021).** A Visual Question Answering (VQA) benchmark for document images, featuring 50K questions on more than 12K images sourced from the UCSF Industry Documents Library. It contains a diverse set of document types (letters, forms, tables, reports), assessing the capabilities of multimodal models in text recognition, document understanding (structure and context), and figure interpretation. The dataset is composed of questions categorized into 9 types, including form-based, table/list-based, layout-based, running text, handwritten text, figure-based, photograph-based, Yes/No, and other.

   **TextVQA ($VQA^T$) Singh et al. (2019).** A VQA benchmark designed to evaluate the ability of MLLMs to read and interpret text within images. It consists of 45,336 questions paired with 28,408 images with a focus on categories that frequently contain text, such as billboards, street signs, and product packaging. Each question-image pair includes ten human-provided ground truth answers. The dataset is designed to necessitate OCR for accurate question answering, as many responses rely on text embedded within the visual scene.

   **ChartQA Masry et al. (2022).** A VQA benchmark for charts, designed to evaluate MLLMs' ability to perform visual and logical reasoning. It consists of 9,608 human-authored questions and 23,111 machine-generated questions based on 20,882 real-world charts collected from diverse sources. The dataset covers bar, line, and pie charts, ensuring a variety of styles and topics. The questions are categorized into data retrieval, visual reasoning, compositional reasoning, and combined visual-compositional reasoning, requiring models to extract data, interpret visual attributes, and perform arithmetic/logical operations.

   **AI2D Kembhavi et al. (2016).** A VQA benchmark for diagram understanding and reasoning, for scientific diagrams. It comprises 5,000 grade-school science diagrams, annotated with 118,000+ labelled constituents and 53,000+ relationships, along with 15,000 multiple-choice questions that test comprehension and reasoning. The dataset covers a diverse set of scientific concepts such as food webs, life cycles, and planetary systems. The dataset is divided into a training set of 4,000 diagram images and a blind test set consisting of 1,000 images.

2. General visual question answering benchmarks:

**VQA$^{v2}$ Goyal et al. (2017).** A VQA benchmark, designed to reduce language biases and assess MLLMs' ability to rely on image understanding. It consists of approximately 1.1 million question-image pairs, with 13 million answers sourced from $\sim 200,000$ images in the COCO dataset. A key feature of VQA$^{v2}$ is its balanced question-image pairs, where each question is linked to two similar images that yield different answers, ensuring that models cannot rely solely on language priors. The dataset covers diverse question types, including yes/no, number, and open-ended questions.

**GQA Hudson & Manning (2019).** A VQA benchmark designed to evaluate visual reasoning capabilities by incorporating scene graph representations of images. It consists of 22 million questions derived from 113,000 real-world images sourced from the Visual Genome dataset Krishna et al. (2017), with each question grounded in a structured scene graph that captures object relationships, attributes, and spatial arrangements. This dataset emphasizes compositional reasoning, linguistic clarity, and reduced bias, ensuring questions are well-formed, diverse, and require multi-step inference. The dataset includes detailed annotations specifying the reasoning steps involved, supporting fine-grained evaluation of model interpretability and robustness. Also, GQA provides balanced question-answer pairs to mitigate dataset biases and includes multiple difficulty levels.

**VizWiz Gurari et al. (2018).** A VQA benchmark designed to address challenges faced by blind individuals in obtaining visual information. It comprises over 31,000 visual questions, where each consists of an image captured by a blind user via a mobile phone and a spoken question that was later transcribed. Each question is paired with 10 crowd-sourced answers, allowing for a robust evaluation of answer variability. Unlike existing VQA datasets, VizWiz introduces real-world complexities such as poor image quality, conversational question phrasing, and a high rate of unanswerable questions ($\sim 28\%$), reflecting the natural uncertainties blind users face.

3. General multimodal benchmarks:

**MMMU Yue et al. (2024).** The Massive Multi-discipline Multimodal Understanding and Reasoning benchmark is a dataset designed to evaluate MLLMs on expert-level perception and reasoning across a diverse set of academic disciplines. It contains 11.5K multimodal questions sourced from college exams, quizzes, and textbooks, covering six broad disciplines including Art & Design, Business, Science, Health & Medicine, Humanities & Social Sciences, and Technology & Engineering, spanning 30 subjects and 183 subfields. The dataset includes 30 heterogeneous image types, such as charts, diagrams, medical scans, sheet music, and chemical structures, to test models' ability to integrate textual and visual information. Unlike existing benchmarks, MMMU emphasizes complex domain-specific reasoning, requiring models to apply advanced subject knowledge to solve problems.

**MMBench Liu et al. (2025).** A bilingual, multimodal benchmark designed to rigorously evaluate MLLMs across 20 fine-grained ability dimensions. It consists of over 3,000 multiple-choice questions, covering a diverse range of tasks including object localization, social reasoning, structuralized image-text understanding, and spatial relationships. MMBench introduces a novel CircularEval strategy to ensure robust evaluation by testing models multiple times with shuffled answer choices. Additionally, it incorporates GPT-4-based choice extraction to handle free-form model predictions, improving evaluation accuracy. We report results on the English subset of MMBench.

**SEED-Bench Li et al. (2024a).** A multimodal benchmark designed to evaluate generative comprehension in MLLMs. It consists of 19,000 multiple-choice questions, making it six times larger than existing benchmarks, with human-annotated ground-truth answers to ensure evaluation accuracy. SEED-Bench spans 12 evaluation dimensions covering both spatial and temporal understanding, incorporating image and video modalities. These dimensions are as follows: scene understanding, instance identity, instance attributes, instance localization, instance counting, spatial relations, instance interaction, visual reasoning, text recognition, action recognition, action prediction, and procedure understanding. We report results on the image set of this dataset.

**POPE Li et al. (2023b).** Polling-based Object Probing Evaluation is an evaluation pipeline designed to systematically evaluate object hallucination in MLLMs. Following the approach of Li et al. (2023b), we evaluate MLLMs with POPE built on the validation set of MSCOCO Lin et al.

(2014). Unlike traditional evaluation methods, POPE formulates object hallucination detection as a binary classification task, prompting MLLMs with Yes/No questions about the presence of specific objects in images. It introduces three distinct negative sampling strategies, including random, popular, and adversarial, to assess the tendency of MLLMs to hallucinate objects based on frequency and co-occurrence patterns in training data. POPE provides a stable, scalable, and fair evaluation framework, making use of both human annotations and automated segmentation tools to extend assessments to unannotated datasets.

**MMVet Yu et al. (2024).** A multimodal benchmark designed to systematically evaluate MLLMs on complex vision-language tasks by assessing their ability to integrate multiple core capabilities. It defines six fundamental vision-language capabilities, including recognition, OCR, knowledge, language generation, spatial awareness, and math, and evaluates models across 16 integrated task categories requiring different capability combinations. MM-Vet contains 200 images and 218 open-ended questions, sourced from various datasets and human annotations. To evaluate open-ended responses, MM-Vet employs a GPT-4-based evaluator, to provide a unified and scalable scoring metric across different answer styles and question types.

**ScienceQA Lu et al. (2022).** A multimodal dataset designed to evaluate multi-step reasoning and interpretability in AI systems through science-based question answering. It comprises 21,208 multiple-choice questions sourced from elementary and high school science curricula, covering natural science, social science, and language science across 26 topics, 127 categories, and 379 skills. This dataset includes rich multimodal contexts, with 48.7% of questions containing images (both diagrams and natural images) and 48.2% containing textual contexts, as well as annotated lectures and explanations to support chain-of-thought (CoT) reasoning. The benchmark assesses whether models can generate explanations alongside answers.

**MathVista Lu et al. (2024b).** A benchmark designed to evaluate the mathematical reasoning capabilities of foundation models in visually complex contexts. It contains 6,141 examples, drawn from 31 existing multimodal datasets and three newly created ones. MathVista spans five primary tasks, figure question answering, geometry problem solving, math word problems, textbook question answering, and visual question answering, covering seven reasoning types including algebraic, arithmetic, geometry, logical, numeric commonsense, scientific, and statistical reasoning. The dataset features diverse visual contexts, such as natural images, geometry diagrams, scientific figures, charts, tables, and function plots.

4. Autonomous Driving benchmark:

**LingoQA Marcu et al. (2024).** A large-scale vision-language dataset designed to evaluate visual question answering in autonomous driving. It consists of 28,000 unique short video scenarios, video captioning, and 419.9K question-answer pairs, covering a broad range of perception and reasoning based tasks. The dataset features free-form questions and answers, extending beyond object detection to include driving behaviour and scenery based questions, while allowing for more robust evaluation with greater answer variability. Question types include a variety of categories that encompass different aspects of understanding the driving task. Below, we present these categories along with an example of each type.

(a) Action identification: *What action are you taking as the current driver?*
(b) Action justification: *Why are you taking this action?*
(c) Object/Scenery identification: *What type of structures do you see on the right side of the road?*
(d) Object/Scenery description: *Can you describe the SUV you see?* or *How is the weather today?*
(e) Attention: *What are you paying attention to?*
(f) Anticipation: *What are you adjusting your position in anticipation of?*
(g) Object localization: *Is the pedestrian crossing the road on a zebra crossing?*
(h) Counting: *How many parked cars can you spot?*
(i) Counterfactual: *How should you react if there are pedestrians still crossing the zebra crossing?*

Table 10: Comparison of various token merging strategies within Tiled MoVE. SA, SI, CC, and CA denote sequence append, sequence interleaving, channel concatenation, and cross-attention, respectively.

| MoVE | Merging | $VQA^T$ | GQA | VizWiz | MMB | POPE | SEED | SQA | MMVeT | Avg. |
|------|---------|---------|-----|--------|-----|------|------|-----|-------|------|
| I + SAM | SA | 63.8 | 64.0 | 54.4 | 68.0 | 87.5 | 69.5 | 72.4 | 34.7 | 64.3 |
| | SI | 65.2 | 64.2 | 54.7 | 68.7 | 87.8 | 69.1 | 74.6 | 35.2 | **64.9** |
| | CC | 63.4 | 63.9 | 54.2 | 67.5 | 87.1 | 70.8 | 71.2 | 34.6 | 64.1 |
| | CA | 62.6 | 63.5 | 53.4 | 66.9 | 86.9 | 68.3 | 71.4 | 34.7 | 63.5 |
| I + ConvNeXt | SA | 63.5 | 63.7 | 53.9 | 66.5 | 86.7 | 68.9 | 71.5 | 34.7 | **63.7** |
| | SI | 64.2 | 63.3 | 54.7 | 64.7 | 86.4 | 68.3 | 73.4 | 34.6 | **63.7** |
| | CC | 62.6 | 63.9 | 54.0 | 66.4 | 87.0 | 68.5 | 71.1 | 34.6 | 63.5 |
| | CA | 62.3 | 63.1 | 53.5 | 63.5 | 85.9 | 67.3 | 70.7 | 33.8 | 62.5 |
| I + DINOv2 | SA | 65.4 | 64.2 | 55.3 | 69.2 | 87.4 | 70.6 | 73.9 | 34.7 | 65.1 |
| | SI | 68.5 | 64.3 | 56.8 | 71.3 | 88.0 | 72.9 | 76.8 | 35.2 | **66.7** |
| | CC | 63.3 | 64.0 | 53.9 | 67.1 | 87.2 | 68.4 | 71.0 | 34.5 | 63.7 |
| | CA | 63.4 | 63.6 | 54.2 | 66.2 | 86.8 | 64.5 | 70.3 | 33.8 | 62.9 |
| S + SAM | SA | 63.6 | 63.5 | 54.3 | 68.0 | 87.4 | 69.1 | 72.5 | 34.8 | 64.2 |
| | SI | 65.0 | 64.3 | 54.5 | 68.3 | 87.4 | 68.4 | 73.9 | 35.1 | **64.6** |
| | CC | 63.4 | 63.2 | 54.0 | 67.3 | 86.9 | 69.8 | 70.4 | 34.8 | 63.7 |
| | CA | 62.6 | 63.4 | 53.2 | 66.3 | 86.4 | 68.1 | 69.8 | 34.5 | 63.0 |
| S + ConvNeXt | SA | 63.4 | 63.8 | 53.5 | 66.3 | 86.5 | 68.9 | 70.9 | 34.6 | 63.5 |
| | SI | 64.5 | 63.6 | 54.7 | 64.6 | 86.8 | 69.0 | 74.1 | 34.6 | **64.0** |
| | CC | 62.3 | 63.8 | 53.6 | 66.5 | 86.9 | 68.4 | 70.8 | 33.9 | 63.3 |
| | CA | 62.1 | 62.8 | 53.2 | 64.6 | 86.1 | 67.2 | 70.2 | 34.0 | 62.5 |
| S + DINOv2 | SA | 65.3 | 64.0 | 55.6 | 68.7 | 87.0 | 70.1 | 73.9 | 34.1 | 64.8 |
| | SI | 68.1 | 64.2 | 56.0 | 70.9 | 88.2 | 71.8 | 76.2 | 35.9 | **66.4** |
| | CC | 63.2 | 64.3 | 53.9 | 66.9 | 87.2 | 70.5 | 70.1 | 34.7 | 63.9 |
| | CA | 62.7 | 63.9 | 53.7 | 64.5 | 86.0 | 66.3 | 69.6 | 33.4 | 62.5 |

To further assess model performance, LingoQA introduces Lingo-Judge, a learned text classifier that achieves a 0.95 Spearman correlation with human ratings of model performance. We calculate Lingo-Judge (Lingo-J), BLUE, METEOR, and CIDEr scores on the evaluation suite of the LingoQA benchmark, which consists of 1,000 additional examples.

## A.2 Experimental Settings

In this section, we describe the detailed experimental settings for our empirical analysis. Starting with Tiled MoVE in Section 3.1 (D1), we build on the baseline introduced by InternViT Chen et al. (2024c). We employ two vision foundation models as first encoders, InternViT Chen et al. (2024c) and SigLIP Zhai et al. (2023), and combine them with three additional encoders selected for their diverse and complementary strengths: SAM Kirillov et al. (2023) for segmentation, ConvNeXt Woo et al. (2023) for OCR-related performance, and DINOv2 Oquab et al. (2024) for representation learning. As in the baseline, we adopt InternLM2-7B-Chat Cai et al. (2024) as the LLM backbone, with a two-layer MLP as the projector. Token merging is performed via channel concatenation with pre-adaptation, i.e., fusion occurs before alignment, and a shared projector is used throughout.

For the next set of experiments (Section 3.2, D2), we retain the same settings as before, including the LLM, projector, and related configurations. We adopt pre-adaptation as the fusion strategy and apply tiling, while varying the token merging strategies.

For the experiments comparing pre- and post-adaptation (Section 3.3, D3), and to ensure a fair comparison, we fix the best-performing settings from the previous stages, namely sequence interleaving and tiling. More-

Table 11: Ablation study on training settings for vision backbones.

| InternViT | SAM | Freeze | VQA$^T$ | GQA | VizWiz | MMB | POPE | SEED | SQA | MMVet | Avg. |
|---|---|---|---|---|---|---|---|---|---|---|---|
| ✓ | ✗ | ✗ | 57.0 | 62.9 | 52.5 | 64.6 | 86.4 | 65.4 | 66.2 | 31.2 | 60.8 |
| ✗ | ✓ | ✗ | 45.2 | 56.4 | 47.5 | 44.7 | 84.2 | 51.3 | 64.0 | 18.2 | 51.4 |
| ✗ | ✓ | ✓ | 49.5 | 58.2 | 50.6 | 48.3 | 85.4 | 54.7 | 65.2 | 19.8 | 53.7 |
| ✓ | ✓ | ✗ | 67.2 | 63.1 | 55.7 | 71.0 | 87.6 | 69.6 | 75.8 | 35.0 | 65.6 |
| ✓ | ✓ | ✓ | 68.8 | 64.8 | 57.9 | 72.9 | 88.0 | 72.2 | 78.5 | 37.2 | **67.5** |

Table 12: Ablation study on various types of fusion strategies and SFT data.

| Model | VQA$^T$ | GQA | VizWiz | MMB | POPE | SEED | SQA | MMVeT | Avg. |
|---|---|---|---|---|---|---|---|---|---|
| LEO | 68.8 | 64.8 | 57.9 | 72.9 | 88.0 | 72.2 | 78.5 | 37.2 | **67.5** |
| w/ pre-adaptation | 65.2 | 64.2 | 54.7 | 68.7 | 87.8 | 69.1 | 74.6 | 35.2 | 64.9 |
| w/ sequence appending | 67.9 | 63.1 | 56.3 | 71.2 | 87.3 | 71.8 | 78.4 | 36.3 | 66.5 |
| w/ channel concatenation | 67.3 | 62.8 | 54.3 | 70.9 | 87.6 | 72.0 | 78.4 | 35.7 | 66.1 |
| w/o tile segmentation | 64.2 | 63.6 | 54.4 | 67.3 | 86.9 | 70.4 | 74.8 | 35.4 | 64.6 |
| w/ 1.8M SFT data | 68.9 | 64.3 | 55.4 | 72.7 | 89.8 | 73.7 | 76.7 | 36.7 | 67.3 |

over, as reported in Tables 1 and 2, combinations with InternViT consistently outperform those with SigLIP. Therefore, we focus on InternViT as the first vision encoder, while adding two additional combinations for comparison: SigLIP Zhai et al. (2023) and CLIP Radford et al. (2021).

## A.3 Benchmark Details

This section provides additional details on results reported in the main text.

Table 10 presents the complete results across all eight benchmarks for the comparison of token merging strategies within Tiled MoVE, as summarized in Table 2. Although sequence interleaving is not always optimal for every vision–language task and MoVE combination, it achieves the highest average score in five out of six combinations.

Table 11 provides detailed results of the ablation study on different training strategies for vision encoders. Unfreezing both vision encoders does not improve performance. This is likely because the pretrained encoders already provide strong visual representations, and fine-tuning them may disrupt their learned features rather than enhancing them.

## A.4 LEO performance beyond the MoVE setting

In addition to the main evaluation on MoVE MLLMs, we compare LEO against representative non-MoVE (general) models with similar resource constraints, i.e., those trained with comparable amounts of data and using a 7B language model (Table 13). For reference only, we also include larger-resource models. The goal here is not to compete with frontier-scale systems Tong et al. (2025); OpenAI (2024); Li et al. (2025); Meta AI (2024); Zhu et al. (2025), which operate with orders of magnitude more resources, but rather to place our approach into perspective against fair baselines. Despite being optimized for the MoVE setting, LEO demonstrates strong multimodal reasoning and understanding, achieving high scores on ChartQA (71.0), DocVQA (80.1), GQA (64.8), POPE (88.0), SQA (78.5), MMVet (37.2), and MMBench (72.9). It also secures the second-best performance on VizWiz, MMMU, AI2D, and SEED. Compared to other general MLLMs that support high-resolution inputs, LEO performs competitively; for instance, it surpasses the LLaVA-NeXt Liu et al. (2024c) model on all available benchmarks and achieves superior results in six out of seven benchmarks when compared to Monkey Li et al. (2024c), underscoring its robust visual reasoning capabilities. Models highlighted in grey are included for reference only, as they use significantly more training data or larger LLMs.

Table 13: Comparison with general MLLMs with similar resource constraints. All models in the lower section use a 7B language model. Models in grey use significantly more training data or larger-scale LLMs.

| Model | ChartQA | DocVQA | VQA$^T$ | GQA | VQA$^{v2}$ | VizWiz | MMB | MMMU | POPE | AI2D | SEED | SQA | MMVet |
|---|---|---|---|---|---|---|---|---|---|---|---|---|---|
| GPT-4V OpenAI (2024) | 78..5 | 88.4 | 78.0 | 36.8 | - | - | 75.8 | 56.8 | - | 78.2 | 69.1 | 75.7 | - |
| Cambrian-1-34B Tong et al. (2025) | 75.6 | 75.5 | 76.7 | 65.8 | - | - | 81.4 | 49.7 | - | 79.7 | 75.3 | 85.6 | - |
| Llama 3.2-11B Meta AI (2024) | 83.4 | 88.4 | - | - | 75.2 | - | - | 50.7 | - | 91.1 | - | - | - |
| Instruct-BLIP Dai et al. (2023) | - | - | 50.1 | 49.2 | - | 34.5 | 36.0 | - | - | - | - | 60.5 | 26.2 |
| LLaVA.1.5 Liu et al. (2024a) | - | - | 58.2 | 62.0 | 78.5 | 50.0 | 64.3 | - | - | - | - | 66.8 | 31.1 |
| InternVL Chen et al. (2024c) | - | - | 57.0 | 62.9 | 79.3 | 52.5 | 64.6 | - | 86.4 | - | 65.4 | 66.2 | 31.2 |
| LLaVA-NeXt Liu et al. (2024c) | - | - | - | - | - | - | 67.4 | 35.8 | 86.5 | - | 70.2 | - | - |
| VILA Lin et al. (2024) | - | - | 64.4 | 62.5 | 79.9 | 57.8 | 68.9 | - | 85.5 | - | 61.1 | 68.2 | 34.9 |
| VILA-1.5 Lin et al. (2024) | 52.7 | 40.6 | 68.5 | - | 83.0 | - | - | 38.6 | - | 76.6 | 73.8 | - | - |
| Monkey Li et al. (2024c) | 65.1 | 66.5 | 67.6 | 60.7 | - | 61.2 | - | - | - | 62.6 | - | 69.4 | - |
| Leo | **71.0** | **80.1** | **68.8** | **64.8** | 78.3 | 57.9 | **72.9** | 36.4 | **88.0** | 69.6 | 72.2 | **78.5** | **37.2** |

## A.5 Additional efficiency analysis

We report the computational characteristics of the four token-merging strategies used in Tiled MoVE. For each method, we measure the number of visual tokens per tile, the inference throughput, and the parameter count excluding the LLM. Table 14 summarizes the results. The four strategies exhibit clear efficiency–accuracy trade-offs. Sequence interleaving achieves the highest average score, indicating that tightly mixing visual tokens produces richer joint representations, though with a slight reduction in throughput. Sequence appending offers a balanced middle ground, providing strong accuracy with marginally higher speed. Channel concatenation is the most efficient option, but consistently yields lower accuracy. In contrast, cross-attention introduces the highest parameter overhead and the slowest inference speed, while also delivering the weakest accuracy among the methods.

Table 14: Efficiency analysis of merging strategies.

| Merging | #Tokens/tile | #Tokens/s | Params. | Avg. |
|---|---|---|---|---|
| SA | 512 | 41.14 | 679M | 64.3 |
| SI | 512 | 40.31 | 679M | 64.9 |
| CC | 256 | 42.49 | 670M | 64.1 |
| CA | 256 | 38.74 | 680M | 63.5 |

## A.6 Does Scaling the Training Data Improve Cross-Attention Merging Performance?

To evaluate whether increasing the amount of supervised training data benefits the cross-attention merging strategy, we train cross-attention variants using the larger Eagle-1.8M dataset Shi et al. (2025). We conduct experiments on two encoder configurations, and report results across eight standard benchmarks in Table 15. Training the cross-attention merging strategy with the larger Eagle-1.8M dataset yields only marginal improvements in both encoder configurations. The average gains range from 0.2-0.3 points, indicating that increasing the supervised training scale has limited impact on the effectiveness of cross-attention fusion.

## A.7 Additional Visualization Results

Figure 5 provides additional visualizations showcasing Leo's capabilities across OCR-related tasks, chart interpretation, and document understanding. In the OCR setting, Leo is able to localize and read fine-grained textual elements embedded within complex scenes. For example, it not only identifies the player wearing a green jersey on the right but also correctly recognizes the number printed on the jersey, demonstrating robust text recognition under challenging visual conditions. Beyond natural images, Leo exhibits strong performance on structured documents: it can navigate dense layouts, focus on the appropriate textual regions, and extract semantically relevant information. In one example, Leo

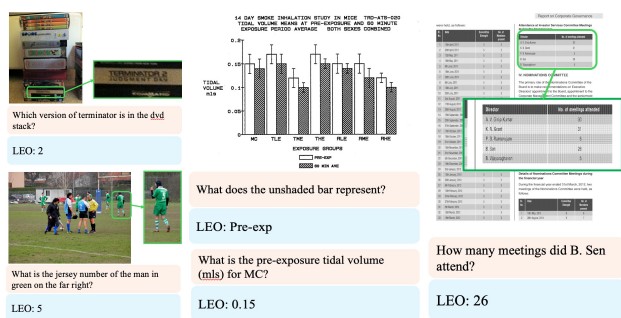

Figure 5: More qualitative results of OCR and chart and document understanding.

determines the number of meetings attended by "*B. Sen*" by precisely locating the corresponding entry and

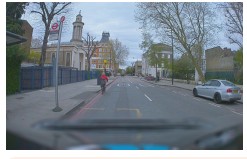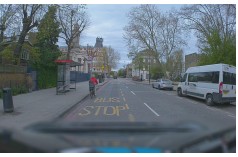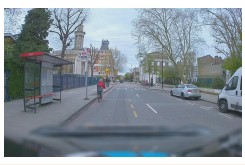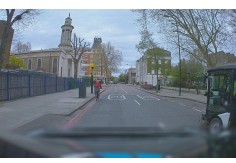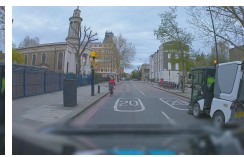

Is it safe for you to increase your speed in the current driving conditions? Provide a rationale.

LEO: No, it is not safe for me to increase my speed in the current driving conditions because there is a cyclist ahead of me, and I need to maintain a safe distance from them.

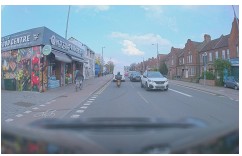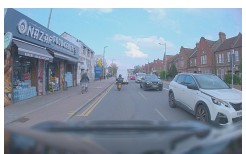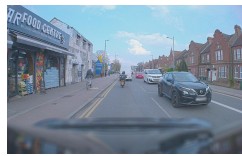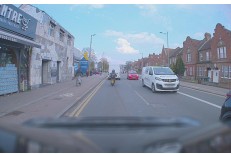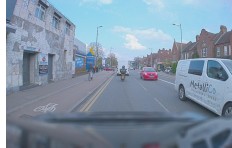

In which direction is the red car driving? Answer the question using one or two sentences.

LEO: The red car is driving in the opposite direction.

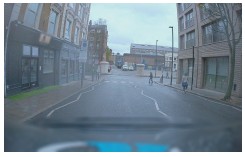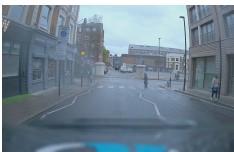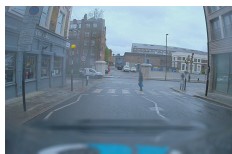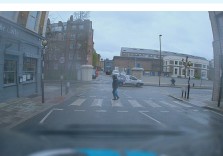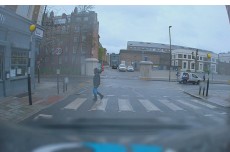

Can you spot any immediate hazard on the road ahead? Answer the question using one or two sentences.

LEO: Yes, there is a zebra crossing ahead with a pedestrian crossing.

Figure 7: Additional qualitative results of scene understanding. The images are taken from LingoQA Marcu et al. (2024)

interpreting the surrounding context. These cases highlight LEO's ability to integrate visual perception and text reasoning, enabling accurate understanding across both scene-text scenarios and complex document analysis.

Figure 7 gives additional qualitative examples of the scene understanding capabilities of LEO in the autonomous driving domain. As shown, LEO can effectively reason about the driving scenario, identify the appropriate safe action, interpret dynamic scenes, and detect any immediate hazards.

Finally, Fig. 6 highlights two representative failure cases of LEO, covering both document understanding and scene-level traffic sign recognition. In the driving scenario, the model correctly detects the circular road marking that indicates a speed limit of 20 mph, yet it misinterprets the question "*Are there any traffic signs?*" by treating the painted road symbol as equivalent to a physical signpost. Since no actual traffic sign is visible in the scene, the model's answer is strictly incorrect, revealing a limitation in distinguishing between road markings and standard signboards. In the second example, which involves document understanding, the model is asked to identify a specific individual associated with a query. LEO incorrectly outputs "*A. Baijal*" after detecting the number four that corresponds to the count of other directorships, rather than linking the correct entity "*H. Khan.*" This error illustrates how LEO can become distracted by salient numerical patterns while failing to ground them in the appropriate semantic context. To-

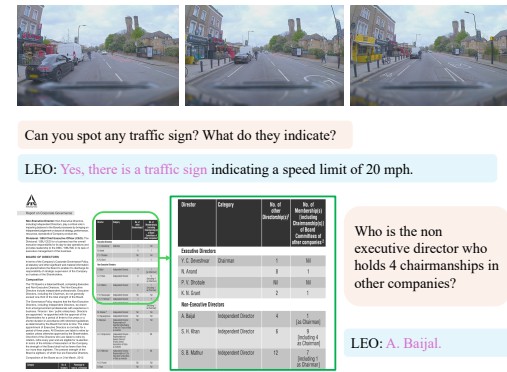

Can you spot any traffic sign? What do they indicate?

LEO: Yes, there is a traffic sign indicating a speed limit of 20 mph.

Who is the non executive director who holds 4 chairmanships in other companies?

LEO: A. Baijal.

Figure 6: Two failure cases of LEO's scene and document understanding.

Table 15: Impact of training data scale on cross-attention merging.

| MoVE | Dataset | VQA$^T$ | GQA | VizWiz | MMB | POPE | SEED | SQA | MMVet | Avg. |
|---|---|---|---|---|---|---|---|---|---|---|
| I + SAM | 1M | 62.6 | 63.5 | 53.4 | 66.9 | 86.9 | 68.3 | 71.4 | 34.7 | 63.5 |
| | 1.8M | 62.5 | 63.6 | 53.6 | 66.9 | 87.1 | 68.5 | 71.2 | 35.9 | **63.7** |
| S + SAM | 1M | 62.6 | 63.4 | 53.2 | 66.3 | 86.4 | 68.1 | 69.8 | 34.5 | 63.0 |
| | 1.8M | 62.7 | 63.7 | 53.0 | 66.5 | 86.9 | 68.6 | 70.4 | 34.8 | **63.3** |

gether, these examples underscore that while Leo demonstrates strong capabilities, it can still conflate visually similar cues or overlook fine-grained semantic relations, leading to incorrect predictions.

