# OpenReview forum: "Rethinking the Mixture of Vision Encoders Paradigm for Enhanced Visual Understanding in Multimodal LLMs"
_TMLR — Accepted by TMLR_

### Review · Reviewer_Q4Tp · 2025-10-29

**Summary Of Contributions:**

Initial vision-language models (VLMs) underperformed on visual tasks relative to our expectations. Recent work has improved the ability of VLMs to see via strategies like mixture of vision encoders (MoVE), image tiling, fusion networks, etc. This paper investigates three design choices of VLMs: (1) combining MoVE and image tiling, (2) visual-token merging, and (3) fusion timing. First, they find that image tiling consistently helps MoVE. Second, they find that sequence appending and sequence interleaving outperform cross-attention and channel concatenation strategies. Third, they find that placing encoder-specific projections before the fusion network outperforms earlier fusion.

**Additional Comments:**

Q1: I find the difference between sequence appending and interleaving interesting. The paper hypotheses that interleaving outperforms appending because it "preserves spatial relationships" via position encoding (since the LLM is causal it can internally encode positions even without position embeddings). Can we add position embeddings to visual tokens to further improve performance or even match the performance of interleaving without interleaving?

Q2: I see that cross-attention tiling performs worse. Since it is the only parametric tile-merging approach, maybe it performs better with more data? Do you see any data-dependent effects?

**Audience:**

Yes

**Audience Explanation:**

VLMs, and multimodal transformers generally, are powerful and popular. The paper investigates subtle strategies to improve performance. The findings are interesting to me, and I suspect many others.

**Broader Impact Concerns:**

no concerns

**Claims And Evidence:**

Yes

**Claims Explanation:**

The submission makes claims based on controlled experiments across 8 vision-language benchmarks. And the findings appear to be consistent across datasets.

**Requested Changes:**

I recommend the paper is accepted regardless, although I have one request:

Please experiment with varying the number of output tokens from the cross-attention tile-merger. If it gives fewer visual tokens to the LLM than the other methods, then maybe this explains the result alone? This can be done, for example, by using single self-attention layer so that the number of input visual tokens matches the output.

---

### Review · Reviewer_Xzjk · 2025-11-12

**Summary Of Contributions:**

This paper investigates Mixture of Vision Encoders for Multimodal Large Language Models (MLLMs) and proposes LEO as a simple yet effective framework for high-resolution visual reasoning. The authors justify their design choices through multiple ablation studies, and benchmark LEO against recent MLLMs and show state-of-the-art performance on multiple VQA and automatic driving datasets.

Overall, this is an excellent paper. The literature review is thorough, and the research question is interesting and practical. The exploration of the design space for Mixture of Vision Encoders for MLLM is well-motivated and well-executed, and the proposed method is simple yet effective. The empirical results are favorable.

**Audience:**

Yes

**Audience Explanation:**

This paper will be of interest to multiple communities within TMLR’s audience, including but not limited to researchers in Multimodal Large Language Models (MLLM), Vision Foundation Models (VFMs), Autonomous Driving, and Multi-teacher Knowledge Distillation.

**Broader Impact Concerns:**

There is no substantial ethical implications beyond average research in MLLMs.

**Claims And Evidence:**

Yes

**Claims Explanation:**

The authors include thorough investigations on multiple key aspects of MoVE design space in Section 3, including dynamic tiling, token merging, and fusion timing. They provide strong evidence for the benefits of tiling, sequence interleaving, and post-adaptation fusion, which they built their framework upon.

**Requested Changes:**

Questions and suggestions:
1. Can Leo be easily scaled up into more than two vision encoders? Currently, LEO takes InternViT-300M and a SAM-L as encoders. Is extensive modification to the current framework needed if you were to scale this up to e.g. InternVL with SAM, DINO V3, and ConvNeXt V2?
2. Can Leo fuse (and benefit from) two MLLMs’ vision encoders, or is it restricted to having only one MLLM within its suite of vision encoders? The paper mentioned "primary encoder" and "secondary expert", but currently it isn’t clear if and how the priority is enforced in practice.
3. The comparison between Leo and RADIO family could be described in more details, for example, RADIO models have adaptors for each teacher on top of the backbone, but it wasn’t clear from 5.4 if RADIO was evaluated with only its backbone. Similarly, was tiling also used for RADIO? At the very least, RADIO is cheaper than LEO at inference time (albeit more expensive to train), which is also unmentioned in the current section.
4. I find Tiling in D1 (subsection 3.1) to be the weakest in terms of empirical support among the three investigation directions. Currently the section reads "tiling is better than no tiling", which is not exactly a new finding. It could be beneficial to compare dynamic tiling to other tiling methods and perhaps even direct representation upsampling methods (AnyUp by Wimmer et al. 2025 might be a good baseline for this).
5. Subsection 3.2 on Token merging introduced four main token merging methods in current literature, however, the computational cost for each was not discussed. Similar question goes for post-adaptation fusion in Subsection 3.3.


**Clarifying 1 and 2 is critical to secure my recommendation, addressing 3-5 will further strengthen this work.**

---

### Review · Reviewer_1wnh · 2025-11-12

**Summary Of Contributions:**

Paper 6111 conducts a systematic study of design choices in Mixture of Vision Encoder (MoVE) based models.
The paper's focus lies on the interaction between visual reasoning, token-level merging and fusion timing. The papers findings are used to devise Leo, a MoVE inspired multi-modal large language model for visual reasoning.
The experimental section presents an extensive set of experiments on OCR and chart understanding, general VQA as well as general multi-modal evaluation benchmarks.  The authors observe competitive or improved results.

**Audience:**

Yes

**Audience Explanation:**

Yes, the paper is well written, its experiments are well connected and the paper focuses on a relevant vision topic.

**Broader Impact Concerns:**

The Broader impact statement covers potential concerns very well.

**Claims And Evidence:**

Yes

**Claims Explanation:**

Yes, very much so! The paper carefully investigates tiling (Table 1), token merging (Table 2) and feature fusion ( Table 3 ).
Based on their analysis the authors follow Chen et al. (2024a) for their tiling approach, choose to work with two independent MLP projections for token merging and finally a tile-level sequence interleaving strategy for feature fusion.

Furthermore Table 4 presents convincing benchmark results for the LEO architecture which is built based on the lesions learned from tables one two and three. The numbers look extremely promising, which further validates the paper's approach.

Last but not least the papers includes an ablation study, efficiency analysis, an application to autonomous driving application as well as visualizations of network outputs.

**Requested Changes:**

None, this paper is ready for publication.

Edit: It would be neat if supplementary code could be added for reproducibility purposes.

---

### Decision · Action_Editor_oPXG · 2026-01-17

**Recommendation:** Accept as is

**Audience:**

Yes

**Audience Explanation:**

All reviewers officially recommend acceptance and agree on the existence of an audience. VLMs and transformers are popular and powerful models that are receiving a lot of attention. A careful study of how exactly to handle the multi-modality of such models is informative and of interest for the continued work on these models. Furthermore the multi-teacher distillation approach and the autonomous driving application provide additional audiences in the TMLR community.

**Claims And Evidence:**

Yes

**Claims Explanation:**

All reviewers officially recommend acceptance and agree the claims fit the evidence. The author response addressed the concerns of 1wnh on code for reproducibility and a missing control experiment of Q4Tp and the questions of Q4Tp too. The submission and its revision provide controlled and consistent experiments across multiple vision and language benchmarks.

- The systematic study is indeed systematic with multiple datasets and ablations.
- The results of this empirical study result in a method, LEO, that is competitive with existing methods while only needing fewer encoders and achieving a lower generation time in fewer parameters.
- Results are shown for vision-language benchmarks and autonomous driving.